

# Debris Flow Risk Mapping Based on GIS and Extenics

Wenbo Xu [1,2], Xueru Zhang[1,2], Yangjuan Zou[1,2], Chunyu Zhang[1,2] and Siyu Liu[1,2]

[1]Center for Information Geoscience, University of Electronic Science and Technology of China
[2]School of Resources and Environment, University of Electronic Science and Technology of China
*Correspondence to*: xuwenbo@uestc.edu.cn; Tel.: 0086-28-61830279; Fax: 0086-28-61831571

**Abstract.** Debris flow, a very dangerous natural disaster, frequently occurs in mountainous areas of Sichuan province. China. Here, we applied the extenics method, which is normally used in single debris flow risk assessment, towards a large-scale debris flow risk assessment for the first time, and built the classical matter elements and joint domain matter elements for assessment of the debris

flow risks in Sichuan province. Eight factors, including relative elevation, slope, rock hardness, rainfall, gully density, vegetation coverage, occurrences of historical debris flow and historical earthquake occurrences were selected for debris flow assessment by using geographic information system technology and weight analysis approach. Based on the risk assessment, the debris flow risk map was generated. Results indicate that areas with high risk and very high risk accounted for 21.32% and 14.35% of the whole province, respectively. 76% of the verification points fall within the moderate, high and very high risk areas, suggesting high accuracy

of extenics method in large-scale assessment areas. Thus, the Geographic Information System (GIS) and extenics based methods could provide a strong support for debris flow management in the region.

## 1 Introduction

Debris flow, which is a very dangerous natural disaster, becomes very common in Sichuan mountainous areas. The quantification of debris flow risks can provide very valuable resources for the prevention and control of regional debris flow study and

management. The debris flow risk in this paper is defined based on the definition given by the United Nations in 1991 for discussing debris flow risks (Zhang, 2012) and is studied from the perspective of natural attributes of debris flow risk.

Due to limitations of data collection and research methods adopted in the early age, the risk assessment study on the debris flow mostly focused on the single ditch and the impact factors were simple, such as topography, and forms of debris flow, rainfall and flood (Eldeenet, 1980; Hollingsworth and Kovacs, 1981). This kind of study requires field survey data and large workload. Since the

1990s, the development and popularization of Remote Sensing (RS) and GIS have enabled the data acquisition to become convenient and provided new approaches for debris flow risk research. Based on RS data, artificial neural network method has been applied to investigate risk assessment of geological disasters (Gomez and Kavzoglu, 2005).

Some scholars adopted the factors including rainfall intensity, topographic characters and soil features, etc. to establish the regression model for risk assessment of debris flow (Friedman and Kavzoglu, 2014; Guzzetti et al., 2008). However, some of the

researches mentioned above only discussed the factors for risk assessment and neglected the contributions of each factors to debris flow, that is no weight analysis. The grey correlation analysis model, Analytic Hierarchy Process (AHP) or combined weight can be applied to analyze the weight of each contributing factor for geological disasters. Such approach demonstrated consistency between assessment results and actual performances (Yang et al., 2011; Zhang et al., 2013; Zhang et al., 2003). Also, some scholars established the debris flow risk model by using the Kalman filtering to predict the occurrence of debris flow, resulting in smaller error compared

with the neural network (Lin et al., 2012; Liu et al., 1993). In the single ditch debris flow risk assessment, the actual measured values of eleven factors including bending coefficient of the main trench and the cutting of the river basin density were quantified for the quality value of the evaluation risk, and the results showed that the integrated judgment method based on mathematical function is effective (Sun and Bi, 1997; Lin et al., 2012). The Extenics theory was mostly used in the single ditch debris flow assessment to resolve the conflicts between fuzziness and uncertainty of debris flow system and objectivity and certainty of debris flow risk

assessment (Gu et al., 2010; Shang et al., 2010).

Currently, RS and GIS have become increasingly mature in the application of debris flow and RS data becomes easier to acquire, with better resolution. Based on previous research results, this paper calculated relative altitude, slope, rock hardness, gully density,



vegetation coverage, rainfall, the occurrence of historical debris flows and the occurrence of historical earthquakes, and then applied the extenics method for the debris flow risk assessment at a large scale for the first time (Xiong et al., 2009; Ning et al., 2013;Lu et al., 2011; Ma et al., 2012; Xu et al., 2015: Xu et al., 2013).

## 2 Data and Methodology

### 2.1 Study area

Study area China Sichuan Province is located in the central part of southwest China and its topography is high in the west and low-lying on the east. The paralleled ridge-valley and the hills of central Sichuan Basin lie in the east of Sichuan, with an elevation ranging from 500 to 2,000m. The Chengdu Plain is in the central part and Western Sichuan Plateau is in the west, whose elevation exceeds 3,000m (Fig. 1). The geomorphology in the Sichuan province is complicated with mountainous as the main features, where there are four different type of land-forms, mountain, hill, plain and plateau, with the percentage of 74.2%, 10.3%, 8.2% and 7.3%,

respectively. The area of Sichuan province is proximately 486,000 square kilometers and lies within 97°21' to108°33' E and 26°03'-34°19' N. The upstream of the Yangtze River passes through the province, and extent of the province is 1,075km from the east to the west and 921km from the north to the south. The climate conditions varies from different regions: for the eastern part, warm in winter, dry in spring, hot in summer, rainy in autumn, with more cloud and mist but less sunshine; and for the western part, much colder, longer winter, having no summer in most time, with sufficient sunshine, concentrated rainfall, distinct separation of dry and

rainy seasons and huge vertical change in climate.

### 2.2 Data

   The data of this paper include influencing factors and historical conditions for the occurrence of debris flow. The earthquake occurrence points in the period of 1970 -2003 were collected. These data are in tabular format and include the following information: time, place, longitude, latitude and magnitude of each earthquake. They were collected from the official website of China Earthquake

Networks Center (CENC). The debris flow occurrence points from 1981 to 2004 were collected from the Research Institute of Geological Environment of China. These data include longitude, latitude, time and place of debris flow, which provide support for model evaluation and data analysis. The daily rainfall data in 1990 -2015 were collected from the official website of China eteorological Administration. 41 stations, their longitude, latitude and places and the rainfall from 20:00 to 20:00 of the following day are included. The other factors that affect debris flow risks were collected, including terrain, landform, geology, vegetation. The

terrain and landform data are mainly derived from the SRTM-DEM (90m*90m), which was used to extract the relative elevation, slope and gully density. Geological data were used to calculate the rock hardness. NDVI data are derived from Computer Network Information Center of Chinese Academy of Sciences, Geospatial Data Cloud and MODND1T product, which was used to calculate the vegetation coverage. All of the products were processed into thematic maps using of ArcMap 10.2. Table 1 shows the details of all these thematic maps.

**2.3Methodology**

   Extenics is a relatively new transversal subject established through the research led by Prof. Wen Cai from Guangdong University of Technology, China, which consists of extension theory, extension method and extension engineering method (Liu and Gu 2010). As a formalized model, extenics can be used to study the extensive possibility of matters and rules and methods of exploitation and innovation, and to solve contradictory problems.

In terms of integrated evaluation of matters by extenics, the objects described or evaluated, their characteristics and magnitude of objects concerning these characteristics are combined as one unit (i.e. matter element) and the correlation function value of extension set or degree of correlation is used to describe the affiliation of characteristic parameters with the objects studied so as to extend the qualitative description of "belonging to" or "not belonging to" as quantitative description(Zhu et al., 2012). The extension method includes five steps, as shown in Fig. 2 below.

(1)Determination of classical domain and joint domain

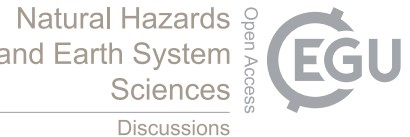



Suppose that there are n debris flow risk assessment indexes, namely $C_1, C_2, C_3, …, C_n$ and the debris flow area is divided into m risk

zoning classes. Thus, the classical matter element model for assessment is expressed as follows:

Let

$$R_j = (N_j, C_i, V_{ji}) = \begin{bmatrix} N_j & C_1 & V_{j1} \\ & C_2 & V_{j2} \\ & M & \\ & C_n & V_{jn} \end{bmatrix} = \begin{bmatrix} N_j & C_1 & \langle a_{j1}, b_{j1} \rangle \\ & C_2 & \langle a_{j2}, b_{j2} \rangle \\ & M & \\ & C_n & \langle a_{jn}, b_{jn} \rangle \end{bmatrix} \qquad (1)$$

where, $N_j$ means Class j(j = 1,2,3,…m)for debris flow risk assessment; $C_j (i = 1,2,3,…,n)$ means assessment factor j for debris

flow risk class $N_j$; $V_{ji}$ means range of magnitude for $N_j$ in relation to $C_j$, or range of data selected for each class in debris flow risk

assessment in relation to relevant assessment factors or classical domain.

Let

$$R_j = (P, C_i, V_p) = \begin{bmatrix} P & C_1 & V_{P1} \\ & C_2 & V_{P2} \\ & M & \\ & C_n & V_{Pn} \end{bmatrix} = \begin{bmatrix} P & C_1 & \langle a_{P1}, b_{P1} \rangle \\ & C_2 & \langle a_{P2}, b_{P2} \rangle \\ & M & \\ & C_n & \langle a_{Pn}, b_{Pn} \rangle \end{bmatrix} \qquad (2)$$

where, P means all classes of debris flow risk assessment; $V_{Pi}$ means range of magnitude for P in relation to $C_i$, or joint domain

of P.

(2)Determination of the matter element to be evaluated

For the matter element P to be evaluated (the evaluation unit in this book is 1KM*1KM), the geographic information data collected

are expressed by matter element R or the matter element to be determined.

$$R = \begin{bmatrix} P & C_1 & v_1 \\ & C_2 & v_2 \\ & M & \\ & C_n & v_n \end{bmatrix} \qquad (3)$$

where, P means magnitude of a specific evaluation unit $V_i$ in relation to evaluation factor $C_i$ or specific data collected for the unit

to be evaluated.

(3)Establishment of correlation functions

In the extension set, a correlation function indicates the degree to which the matter has a certain property.

Suppose that the distance of P and $N_j(j = 1,2,3,…,m)$ in relation to evaluation index $C_i$ is $\rho(x_i, x_{ji})$; the distance of P and

$N_P$ in relation to $C_i$ is $\rho(x_i, x_{pi})$, then the correlation function of the evaluation index $C_i$ of the unit to be evaluated in relation to

pre-warning class j is expressed as follows:

$$K_j(x_i) = \rho(x_i, x_{ji})/(\rho(x_i, x_{Pi}) - \rho(x_P, x_{ji})) \qquad (4)$$

And the correlation function must match practical problems. Thus, debris flow factors are divided into three classes, A, B and

C(Zhang, 2013).

①Class A index:

For the evaluation index whose magnitude is consistent with or opposite from the trend of debris flow risk degree (or the minimal

point is in the central point of the interval), the junior correlation function based on distance is adopted.

$$\rho(x_i, x_{pi}) = 2\left[\left|x - \frac{a+b}{2}\right| - \frac{1}{2}(b - a)\right] = \begin{cases} 2(a - x) & a \le x < (a + b)/2 \\ 2(x - b) & (a + b)/2 \le x \le b \\ 0 & others \end{cases} \qquad (5)$$

②Class B index:

For the evaluation index whose highest risk degree appears on the left of the central point of the value domain interval (or the

minimal point is on the left of the interval), the junior correlation function based on the left distance is adopted.

$$\rho(x_i, x_{pi}) = \begin{cases} x - b & a \le x \le b \\ 0 & others \end{cases} \qquad (6)$$

③Class C index

For the evaluation index whose highest risk degree appears on the right of the central point of the value domain interval (or the

minimal point is on the right of the interval), the junior correlation function based on the right distance is adopted.




$$\rho(x_i, x_{pi}) = \begin{cases} a - x & a \leq x \leq b \\ 0 & others \end{cases} \tag{7}$$

(4)Weight of assessment factor determination

Here, we use the Analytic Hierarchy Process (AHP) method to determine the weight of factors. There are four major steps described as follows(Bai et al., 2009; Liang, 2007):

①Establishing a hierarchical structure model.

The structure model is normally composed of three or more layers: The upmost layer is called target layer, which is the target to be achieved in the system. It generally composed of one index. The intermediate layer is the criterion layer and it is normally composed of two or more secondary indexes, which are mainly controlled by the target layer and the major impact indexes for the purpose of the system. When the number of the factors is larger than 9, the criterion layer will be decomposed into several sub-criterion layers. The lowest layer is called object layer and normally composed of three or more basic level indexes. These indexes are mostly derived

from secondary indexes and are controlled by secondary target indexes. The basic level indexes are the most immediate factors in the evaluation system. The hierarchical structure model is expressed in Fig. 3.

②Constructing a judgment matrix.

Equation 8 shows a judgment matrix, where $u_i$、$u_j(i, j = 1,2, \dots, n)$ represent impact factors. $u_{ij}$ means the value of relative importance of $u_i$ in relation to $u_j$. $u_{ij}$ is used to construct a judgment matrix U.


$$U = \begin{bmatrix} u_{11} & u_{12} & \dots & u_{1n} \\ u_{21} & u_{11} & \dots & u_{2n} \\ M & M & & M \\ u_{n1} & u_{11} & \dots & u_{nn} \end{bmatrix} \tag{8}$$

We used the consistent matrix method proposed by Santy et al. to determine the importance of individual factors in each layer. This method uses numbers 1 to 9 to judge the ratio between two factors. The importance of every two factors is listed in Table 2.

③Calculating the importance sorting

According to the judgment matrix, the eigenvector ω which corresponds to the maximum eigenvalue $\lambda_{max}$ was calculated. The

equation is expressed as follows:

$$U\varpi = \lambda_{max}\varpi \tag{9}$$

The normalized value of eigenvector $\varpi$ is the sequence of the importance for the assessment factors. That is so called weight distribution.

④Consistency testing

Consistency testing should be conducted on the judgment matrix to ensure rationality of the weight calculated by the equation 10. The testing equation is expressed as follows:

$$CR = CI/RI \tag{10}$$

where, CR means the random consistency ratio of judgment matrix; CI means normal consistency index of judgment index. And CI is obtained according to the equation 11:


$$CI = (\lambda_{max} - n)/(n - 1) \tag{11}$$

RI means average random consistency index of judgment matrix. The RI values for judgment matrix of Orders 1 to 9 are shown in Table 3.

When the value of CR for the judgment matrix U is smaller than 0.1, it is believed that the consistency of U is satisfactory. Otherwise, the factors of U should be adjusted so as to obtain a satisfactory consistency.

(5)Calculating the risk value

$$K_j(p) = \sum_{i=1}^{n} \lambda_i K_j(v_i) \tag{12}$$

Where $K_j(v_i)$ means the factor correlative value and $\lambda_i$ means the factor weight.



## 3 Results and Discussion

### 3.1 Extenics-based Factor Analysis

In this paper, we chose Sichuan Province as our study area. Eight factors were selected as debris flow risk assessment indexes, including relative elevation, slope, rock hardness, rainfall, gully density, vegetation coverage, and the occurrence numbers of historical debris flow and earthquake. The assessment unit is a 1KM*1KM grid. Five classes are divided for the assessment, which are very low risk (I), low risk (II), moderate risk (III), high risk (IV) and very high risk (V). The relationship between risk level and evaluation indexes of debris flow in Sichuan can be obtained by reference to the results of previous researches, as shown in Table 4.

Notes: $C_1 - C_8$ represent the relative elevation, slope, rock hardness, rainfall, gully density, vegetation coverage, the number of historical debris flow occurrences and the number of historical earthquake occurrences, respectively.

The classical matter elements and joint domain matter elements for debris flow risk classes can be obtained from Table 4. Relative elevation represents the difference between the altitude at the highest point and that at the lowest point, which reflects the capacity of debris flow in the valley to carry solid matters and its potential energy. Generally, larger relative elevation of the valley means poorer stability of the slope and faster confluence speed, which generates more energy for the occurrence of debris flow. Table 4 shows relative elevation differs in the whole region and the high relative elevation causes high risk level. In our study area, the relative elevation that ranges from 1,200m to 1,800m will have higher risk level, and that ranges from 1,800m to 2,400m will have lower risk level. For the combined changes, types A and B coefficients can be combined to indicate correlation. Figure 4(a) also indicates that hills dominate in the eastern part of Sichuan, with a smaller relative elevation, thus meaning lower correlation.

Slope means the angle between the slope surface and the plane. Slope does not only affect the stress distribution of mountain slope but also produce a decisive effect on the bulking thickness of loose soil on the slope. The slopes in different areas may constrain the development of debris flow. For the valley with smaller slope, including key area, smaller shear force means larger stability and lower probability for occurrence of debris flow. However, larger slope does not necessarily mean more prone to debris flow. Actually, debris flow occurs within a certain range of slopes. Table 4 shows that debris flow has the largest probability when the slope is around 15℃, in which Types A and B coefficients may be combined to indicate correlation. Figure 4(b) shows a lower correlation of the middle and eastern areas in the region of interest.

Rock hardness reflects the lithology of a valley. For harder rocks in the valley, the solid matters in the debris flow are mainly produced from collapse, with larger grain size of solid matters, longer accumulation time and lower probability for outbreak of debris flow. While for softer rockers in the valley, solid matters are mainly produced from landslide, with smaller grain size of solid matters, shorter accumulation time and higher probability for outbreak of debris flow. It can be reflected from Table 4 that debris flow is more prone with the increase in rock hardness when the hardness lies within 4 and 8 and less prone when within 8 and 14. In such case, Type B and C coefficients may be combined to indicate correlation. It can be shown from Fig. 4(c) that Panxi area in the south has complex rock hardness as the area has varying lithological distribution, ranging from solid basalt and quartzite, moderately solid dolomite rock and calcareous rock, to weakly solid phyllite and tuff.

It is quite obvious that debris flow is more likely to occur in the valleys with more rainfall (Table 4). Thus, Type C coefficients may be used to indicate correlation. Rainfall may provide dynamic conditions and water sources for occurrence of debris flow. Under certain conditions of solid matters, terrain and landform, the time, place and scale of debris flow depend on the rainfall. As rainfall is a dynamic factor, we may choose a proper scale for calculation according to the requirements of assessment. In this paper, the interpolation result of rainfall obtained on July 31, 2009 was used to calculate rainfall correlation. It can be seen from Fig. 4(d) that the middle part of the region of interest has the largest rainfall and also the highest correlation.

Gully density does not only reflect the degree of ground breakage but also the route of debris flow running. The nearer the gully is from the river channel, the more sufficient the freeing conditions will be. Thus, river cutting degree plays an important role in controlling the development of debris flow. In the areas with more main gullies, ground is more seriously broken, leading to more loose trivial matters. Additionally, the catchment area is larger, which generates sufficient power of water sources and appears more likely to cause debris flow (Table 4). Thus, Type C coefficients are used to indicate the correlation of gully density. It can be shown from Fig. 4(e) that gully density is of even distribution in the whole province.



The changes of vegetation coverage are affected not only by natural environment but also by human activities. Also, the growth of vegetation may be impacted by climate and terrain conditions. Human activities may cause damage to vegetation and also lower coverage and larger probability of various types of geological disasters. The vegetation index NDVI is the best index for measuring the growth of plants and spatial density of vegetation. NDVI was used to calculate the vegetation coverage and then used to analyze the relationship between debris flow and vegetation coverage. It can be shown from Table 4 that debris flow is more likely to occur

with the increase in vegetation coverage within the range of 0 to 0.7 and less likely within the range of 0.7 to 1. Thus, Types B and C may be combined to indicate correlation. Figure 4(f) shows relatively low coverage of vegetation in the northern part of Sichuan.

    More historical debris occurrence points indicate more sufficient bad geological conditions and higher risks, which may facilitate the reoccurrence of debris flow. And solid matters were added to debris flow after earthquake. Thus, the number of historical earthquake occurrences is also one of the factors for the occurrence of debris flow. It can be shown from Table 4 that the more the

historical disaster occurrence points are, the more likely the debris flow will occur. Thus, Type C factors may be used to calculate the correlation of number of historical debris flow occurrences and number of historical earthquake occurrences. It can be seen from Fig. 4(g) that the southern part of Sichuan has high correlation of the number of historical debris flow occurrences and from Fig. 4(h) that the eastern part has low correlation of the number of historical earthquake occurrences.

    Secondly, the contributions of all factors are different for debris flow risks. Therefore, the weight of each factor in the debris flow

risk zoning should be determined.

    ①The multi-layer assessment system is built according to the AHP method. This method divides the impact factors into two layers, in which the first layer includes terrain, landform, geology, hydrology, vegetation and historical conditions. Each factor includes multiple sub-factors, as shown in Fig.5.

    Two layers of indexes were sorted according to their importance. Eight factors including relative elevation, slope, rock hardness,

rainfall, drainage density, vegetation coverage, the occurrence number of historical debris flow and earthquake correspond to the overall weight of risk assessment. Then, these indexes were sorted according to weight coefficients, the result of which is shown in Table 5.

    Then, $\lambda_8 = [0.34, 0.17, 0.05, 0.19, 0.05, 0.15, 0.04, 0.01]$.

    The method of extenics was used to assess the risk of debris flow according to Equation (10). Weight was multiplied by eight

factors for superposition of correlation layers. Then, the natural break method commonly used in statistics was applied to further divide risks into five degrees, as shown in Table 6.

### 3.2 Result Verification and Analysis

    The final categorized layer was finally obtained according to Table 7, as shown in Fig. 6.

    We used the historical data that is the debris flow occurrence points to verify the assessment results. The disaster points located in

the very low, low, moderate, high and very high risk areas were 10%, 12%, 20%, 27% and 29%, respectively and the total percentage of the areas with moderate, high and very high risk was 76%. Results indicated that the application of extenics is feasible in large-scale debris flow risk assessment.

    The ArcGIS statistical analysis was adopted to calculate all risk areas and their percentage so as to facilitate our further analysis. The specific data are shown in the following Table7.

From Table 7, the area of the Class I risk is 109,876km2, accounting for 23% of the entire area of Sichuan province. It can be seen that the area is mainly located in Nanchong and west of Bazhong in Sichuan province, with better natural environment and less bad geological conditions.

    The Class II risk area is 115,873 km2, 25% of the whole area of Sichuan. The area is mainly distributed in Chengdu, Deyang and Zigong in the east of Sichuan. Some parts of Aba Prefecture in the north of Sichuan also belong to Class II risk area. These areas have

flat terrain and small height difference as they lie on Hongyuan County and Ruoergai Grasslands, with no objective conditions for occurrence of debris flow, proving little risk.

    The Class III risk area is 79,356km2, 17% of the whole area of Sichuan Province. The area is mainly located inGanzi, Aba and Liangshan Prefectures in the west of Sichuan. Debris flow disasters once occurred in the area. However, debris flow does not



frequently occur in these areas, with moderate relative elevation and moderate vegetation coverage.

      The Class IV risk area is about 101,308km2, 21% of the whole area of Sichuan Province. The area is mainly located in the north of Sichuan, Aba Prefecture, Ganzi in the west and Liangshan Prefecture in the south. These areas have frequent debris flow occurrence, more mountains and relatively large height difference in valleys. Thus, bad geological activities are prone.

      The Class V risk area is about 67,932km2, 14% of the whole area of Sichuan Province. It indicates that the area with the largest

probability of debris flow is the smallest in the whole province. The area is mainly distributed in the south of Liangshan Prefecture (north of Sichuan) and Leshan. It features plenty of bad geology, large relative elevation and abundant rainfall. Thus, debris flow is mostly likely to occur here.

## 4 Conclusions

      The extenics model represents one of the most effective methods for large-scale debris flow risk areas. Compared with the

traditional information value model, this method not only has a high objectivity but also can deal with experts experiences. This paper selected eight factors including relative elevation, slope, rock hardness, rainfall, gully density, vegetation coverage, historical debris flow occurrences and historical earthquake occurrences to study the degree of debris flow risks. The model was applied in the extension model and plenty of correlation functions were established. Also, AHP was adopted to obtain weight and make the analysis of all factors more specific.

20       Before the calculation of correlation function, the relationship between coefficients and risk degrees was determined first (Table 4). We found that the relationship between each factor and risk degree does not change in a single manner. In order to solve the problem, a number of correlation functions in the methods of extenics may be used to express the relationship between factors and debris flow occurrence. The final results show the positions of debris flow risk areas of Sichuan. And the very high risk areas are located in the northwest, some parts of north and south of Sichuan, which match the high-incidence areas of the province.

25       By comparative analysis on the positions of actual debris flow occurrence and risk level area map, the moderate, high and very high risk areas that may pose threat to mankind occupy 52.38% of the region.

      It can be seen from Fig. 6 that in terms of terrain conditions, debris flow is mainly distributed in the deeply cut high and moderately high mountainous areas and the mountains on the edge of basin, such as Aba County of Aba Tibetan Autonomous Prefecture in the north of Sichuan and Shiqu, Seda, Batang and Kangding of GanziYi Autonomous Prefecture in the northwest of Sichuan for Class V

risk areas; and Li County on the south of Aba for Class IV risk areas. These areas are characterized by high mountains, deep gullies, steep terrain and large vertical slope of gully bed, in which some valleys favor the collection of rain water. In terms of geological conditions, debris flow is mainly distributed in the areas with complicated geological structures, developed fracture, serious fold and active movement of new structures. Mountains and plateaus dominate in the western part of the region, most of which are around 3,000m in altitude. With high mountains and steep slopes, the area is very cold and characterized by crushed rocks, serious weathering,

thick loose materials and high seismic intensity. For example, Tianquan County and Hongya County in Ya'an of middle part of Sichuan for Class V risk areas. These areas have loose and weak lithological structures, appear to be easily broken by wreathing, suffer from bad geological conditions such as landslide and collapse, and enjoy rich sources for solid matters in debris flow. In terms of hydrological conditions, debris flow is mainly distributed in rainstorm areas. Rainfall can reduce the friction between loose trivial matters and become the basic power to carry bulk solid matters. These areas are represented by Tianquan County and Hongya County

in Ya'an, as well as Yanyuan County of Ganzi Yi Autonomous Prefecture in south of Sichuan.

      Among eight factors adopted in this study, rainfall varies greatly within a short period. For example, the rainfall lasting different days differ. Other methods may be used to predict the rainfall in the future. Then, the expansion model may be used to give a rapid warning of small-scale debris flow, which will become the orientation for our future research.

### Acknowledgements

45       This work was supported by grants from National Natural Science Foundation of China (41371398).



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

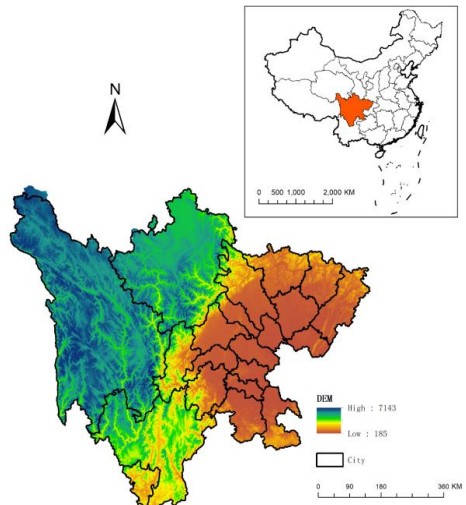

Fig. 1: The location of study area

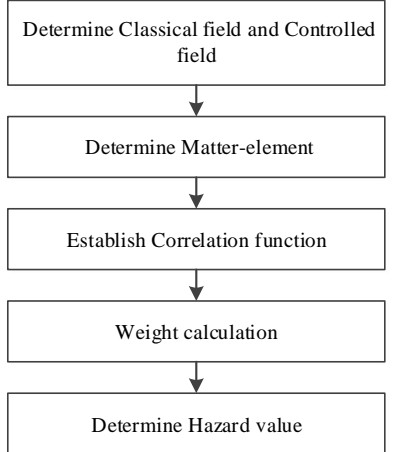

Fig. 2: Extenics evaluation process



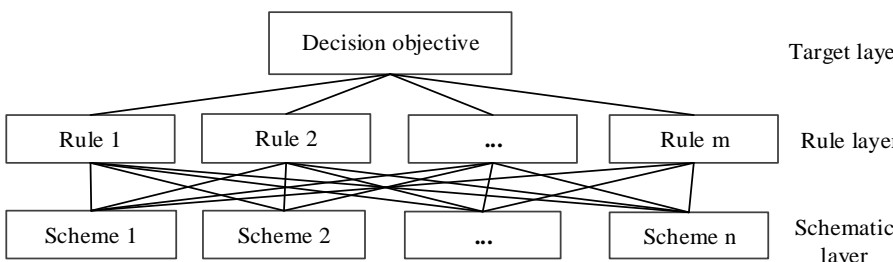

Fig. 3: Hierarchical structure model

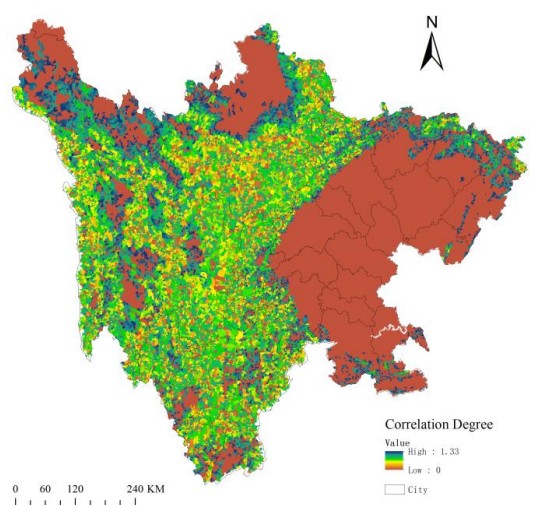

Fig. 4(a): Relative elevation calculation result

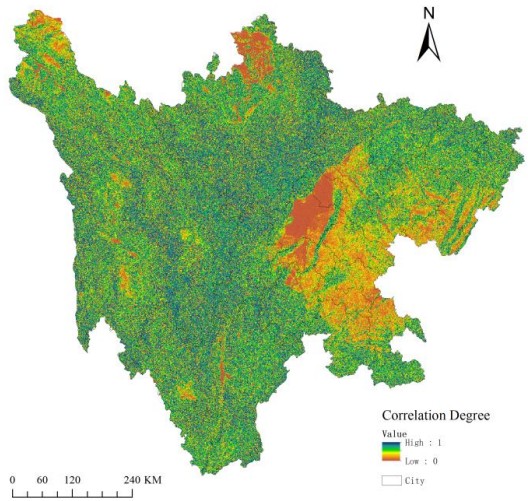

Fig. 4(b): Slope correlation value of Sichuan





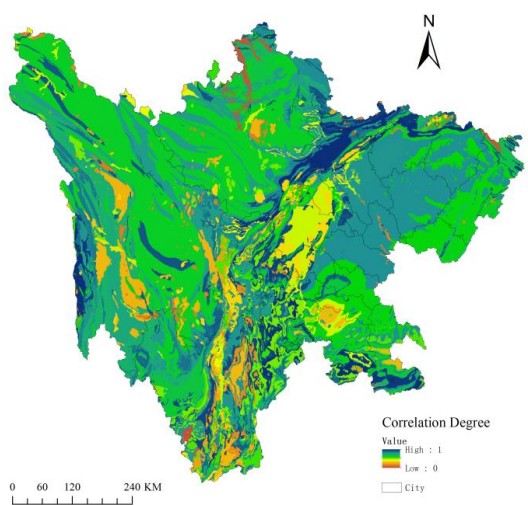

Fig. 4(c): Rock hardness correlation of Sichuan

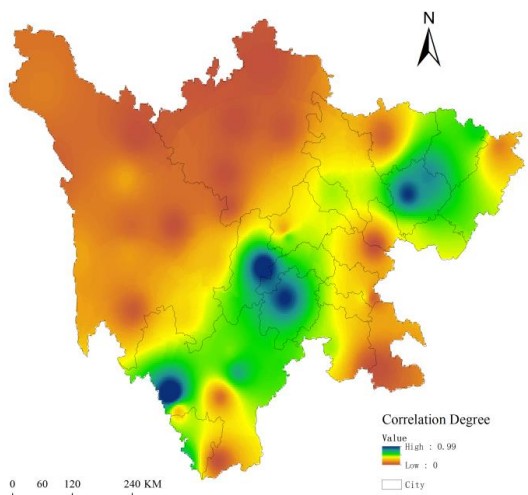

Fig. 4(d): Rainfall correlation of Sichuan



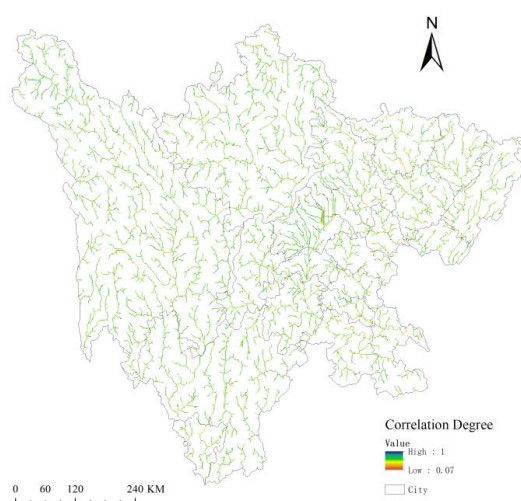

Fig. 4(e): Gully density correlation of Sichuan

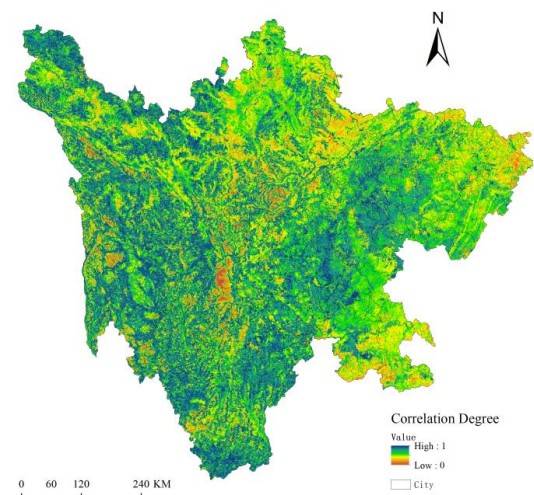

Fig. 4(f): Vegetation coverage correlation of Sichuan



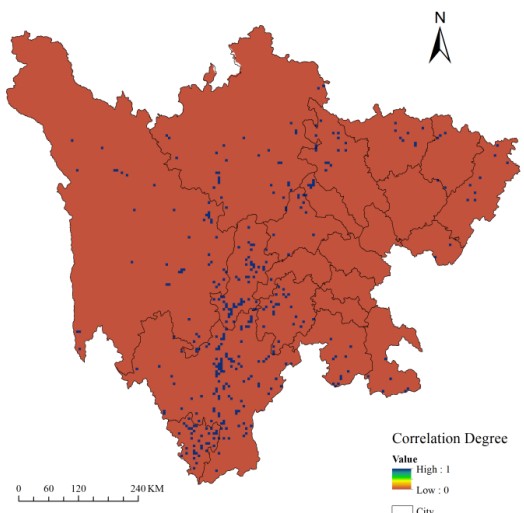

Fig. 4(g): Correlation of the debris flow attribute

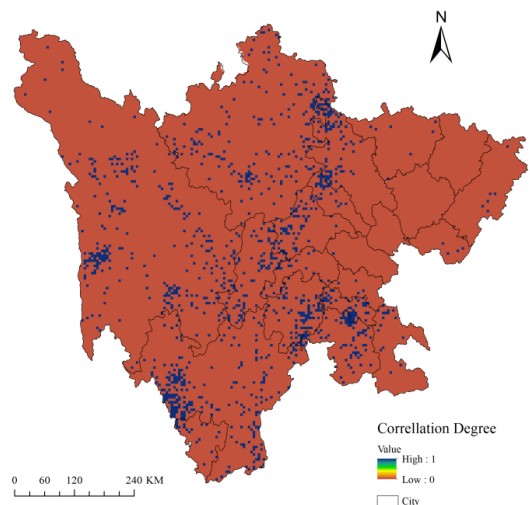

Fig. 4(h): Correlation of the earthquake attribute





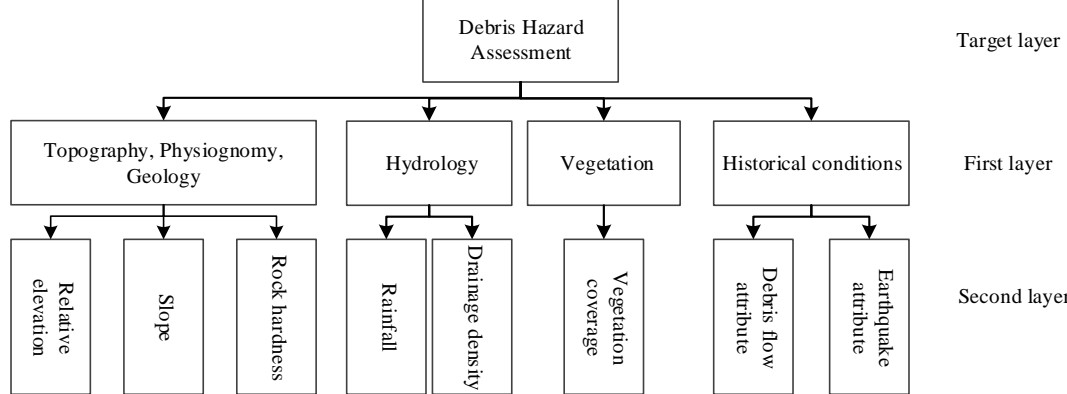

Fig. 5: Multi-layer debris flow assessment system

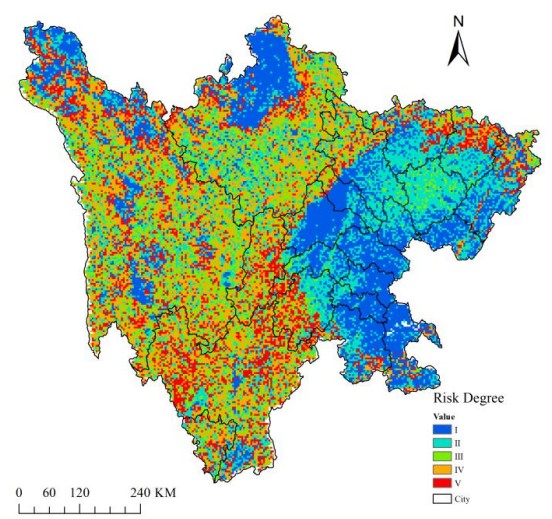

Fig. 6: Division of debris flow risks in Sichuan Province

Table 1: The thematic maps that were used in the prediction of debris flow

|  | Layers | Type | Data scale |
|---|---|---|---|
| Debris flow inventory | Debris flow attribute | Vector(point) layer | 1:1,000,000 |
| Earthquake inventory | Earthquake attribute | Vector(point) layer | 1:1,000,000 |
| Rainfall | Rainfall | Raster | 1km*1km |
| SRTM DEM | Relative elevation | Raster | 1km*1km |
|  | Slope | Raster | 1km*1km |
|  | Gully density | Raster | 1km*1km |
| MODND1T | Vegetation coverage | Raster | 1km*1km |
| Formation lithology | Rock hardness | Raster | 1km*1km |





Table 2: Assessment Factor Importance Scale

| Scale | Meaning |
|---|---|
| 1 | Compared to two factors, the same important |
| 3 | Compared to two factors, one slightly more important than the other |
| 5 | Compared to two factors, one significantly more important than the other |
| 7 | Compared to two factors,one than another very important |
| 9 | Compared to two factors,One is extremely important than the other |
| 2,4,6,8 | Median of the adjacent judgments |
| Reciprocal | The reciprocal of the comparison results |

Table 3: Values of average random consistency indexes RI

| N | 1 | 2 | 3 | 4 | 5 | 6 | 7 | 8 | 9 |
|---|---|---|---|---|---|---|---|---|---|
| RI | 0 | 0 | 0.58 | 0.90 | 1.12 | 1.24 | 1.32 | 1.41 | 1.45 |

Table 4: Relationship between risk level and evaluation indexes of debris flow in Sichuan

| Risk Level | $C_1$(m) | $C_2$(°) | $C_3$ | $C_4$(0.1mm) | $C_5$ | $C_6$(%) | $C_7$ | $C_8$ |
|---|---|---|---|---|---|---|---|---|
| I | >2400 | >45 | 12~14 | 0~100 | 0~0.4 | 0~0.2 | 0 | 0 |
| II | 1800~2400 | 35~45 | 3.9~6 | 100~250 | 0.40~0.55 | 0.2~0.4 | 1~2 | 1~2 |
| III | 0~600 | 25~35 | 10~12 | 250~500 | 0.55~0.70 | 0.4~0.6 | 3~4 | 3~4 |
| IV | 1200~1800 | 15~25 | 8~10 | 500~1000 | 0.70~0.85 | 0.8~1 | 4~5 | 4~5 |
| V | 600~1200 | 0~15 | 6~8 | >1000 | >0.85 | 0.6~0.8 | >5 | >5 |

Table 5: Overall weights of coefficients

| Influence factor | Weight | Order |
|---|---|---|
| Relative elevation | 0.34 | 1 |
| Slope | 0.17 | 3 |
| Rock hardness | 0.05 | 5 |
| Rainfall | 0.19 | 2 |
| Gully density | 0.05 | 5 |
| Vegetation coverage | 0.15 | 4 |
| Debris flow attribute | 0.04 | 6 |
| Earthquake attribute | 0.01 | 7 |

Table 6: Scale of debris flow risks

| Hazard Level | I | II | III | IV | V |
|---|---|---|---|---|---|
| Hazard assessment value | 0.03~0.26 | 0.26~0.39 | 0.39~0.52 | 0.52~0.64 | 0.64~1 |

Table 7: Statistics of Debris Flow Risk Zoning in Sichuan Province

| Risk Zoning Degree | Area (km²) |
|---|---|
| I | 109876 |
| II | 115873 |
| III | 79356 |
| IV | 101308 |
| V | 67932 |