# Peer review of "Debris Flow Risk Mapping Based on GIS and Extenics"

_Natural Hazards and Earth System Sciences, 2018_

## Referee Comment (RC1) · Anonymous Referee #1 · 28 Jul 2018

The paper entitled "Debris Flow Risk Mapping Based on GIS and Extenics" describes an approach to classify regions with high debris flow probability in China.

Even though the the basic idea of large scale hazard indication mapping is very interesting and potenitally benefitting for many applications, the state of the paper does not allow for a decent understanding of all methods applied. Therefore I recommend a rejection of this paper.

My main arguments coming to this conclusions are the following:

- The concept of Extenics is not clear at all. It is not clear why this concept should bring benefit compared to simple correlation of the parameters. As this paper is based very much on the Extenics concept, it has to be described very clearly and carefully and it

has to be explained why it is so important for debris flow hazard mapping. It should also be compared to other, more common approaches. Right now it is only described with catchy keyword such as "innovation", "matter" , "solve contradictionary problems".

- The basic input information, relative elevation, slope, rock hardness, rainfall, gully density, vegetation coverage, historical debris flow and earthquake activity are very essential for the algorithm. But these datasets are not well described at all. A discussion on the quality and uncertainty of these crucial parameters is missing.

- The authors use a gris of 1 by 1 km spatial resolution to derive parameters sich as slope angle and elevation difference. This is way too coarse to derive these parameters for meaningful debris flow hazard detection in mountainous environments.

- The validation with points where debris flows were observed is poor. First of all, as far I understand, is the same data used as input for the model. So it could to be used for validation of the model. Secondly, nearly 50% of all recorded debris flow events fall into the classes very low, low and moderate risk for debris flow. This does not look like a very good model performance as the authors claim.

---

## Author Comment (AC1) · 14 Sep 2018

Dear reviewer,

Thank you very much for your attention and the referee's evaluation and comments on our manuscript "Debris Flow Risk Mapping Based on GIS and Extenics".

We have studied the valuable comments, and tried our best to revise the manuscript. The respond to the reviewer's comments are as follows:

Comment 1: The concept of Extenics is not clear at all. It is not clear why this concept should bring benefit compared to simple correlation of the parameters. As this paper is based very much on the Extenics concept, it has to be described very clearly and carefully and it has to be explained why it is so important for debris flow hazard mapping. It should also be compared to other, more common approaches. Right now it is only described with catchy keyword such as "innovation", "matter", "solve contradictionary problems".

Reply: According to the reviewer's comment, we will compare and analyze the evaluation results of existing debris flow risk assessment methods in first section, the third paragraph. We will make clear statement about the advantages and disadvantages on the existing assessment methods, and give the summary about the difference between extenics approach and other methods on evaluating debris flow risk.

Comment 2: The basic input information, relative elevation, slope, rock hardness, rainfall, gully density, vegetation coverage, historical debris flow and earthquake activity are very essential for the algorithm. But these datasets are not well described at all. A discussion on the quality and uncertainty of these crucial parameters is missing.

Reply: According to the reviewer's comment, we will add a detailed description of the different factor datasets in section 2.2, and discuss the accuracy of the dataset. And meanwhile, giving the reason why the dataset can be chosen.

Comment 3: The authors use a gris of 1 by 1 km spatial resolution to derive parameters sich as slope angle and elevation difference. This is way too coarse to derive these parameters for meaningful debris flow hazard detection in mountainous environments.

Reply: About the reviewer's comment, the author can't understand the reviewer's question clearly. Whether the data is too coarse or too rough. Please make the clear statement.

Comment 4: The validation with points where debris flows were observed is poor. First of all, as far I understand, is the same data used as input for the model. So it could to be used for validation of the model. Secondly, nearly 50% of all recorded debris flow events fall into the classes very low, low and moderate risk for debris flow. This does not look like a very good model performance as the authors claim.

Reply: About the reviewer's comment, the proportion of historical mudslides falling into the corresponding medium, high and heavy danger zones is 76%, rather than 50%. The 50% refers to the proportion of the middle, high and heavy areas corresponding to the study area, which exists ambiguity in this paper. Through highlighting the proportion of historical mudslides falling into the corresponding medium, high and heavy danger zones, the definition can be clear.

---

## Referee Comment (RC2) · Anonymous Referee #2 · 19 Nov 2018

This paper attempts to compute the influence of several factors on Debris Flow risk in a region of China, and to categorize these risks in five different classes.

The idea to analyze the impact of natural and historical factors on debris flows risk, and to assign a weight to all of them might be really interesting and useful. However, the central point of this paper is the use of a new theoretical method (extenics method) which is not explain in a comprehensible manner. Therefore I recommend a rejection of this paper.

Here are my main arguments for this:

1. The methodology to compute correlation factors and weights is not defined in an understandable way. The general concepts of extenics methods (and its advantages

over other methods) has to be defined more precisely and the assossiated mathematical description has to be improved. The symbols used in the equations are not all defined and when they are, it is often without any explanation. Furthermore, the mathematical operation are not explained. For example in eq. 1, M, aj1-ajn and bj1-bjn are not defined, as the operation <a,b> and finally what does exactly means this kind of matrix [] (with columns of different size). Thus, this section is very confused and almost impossible to understand, that takes all credibility away from the results.

2. The input variables ( for instance the historical data page 6, line 34 or rainfall page 7, line 11) are used to verify the data obtained. It is absolutely mandatory to find other parameters (or methods) than the input ones to check the results. Otherwise, they do not have any scientific validity. Moreover, to ensure the validity of the Class V, you claimed that the area (covered by the Class V) is constituted by (among other things) large relative elevation. However, when one looks at the debris flow risks as a function of the relative elevation, one sees that terrains with largest relative elevation fall in Class I and II. As far as I understand, there is a big conmtradiction here.

3. Some of your results are not intuitive, in this case a comment would be helpful. For example, risks become smaller with larger slope (page 5, lines 24-25). In this case, it is written: "For the valley with smaller slope, [...], smaller shear force means larger stability and lower probability of debris flow". It seems for me that your results are in complete disagreement with your interpretation, therefore an explication is expected and needed.

4. Your study is based on field and historical data, which are not well described in the paper. It misses especially a discussion about the quality of the data and the associated uncertainty of measurement.
* * *

---

## Author Comment (AC2) · 25 Dec 2018

Dear reviewer,

Thank you very much for your attention and the referee's evaluation and comments on our manuscript "Debris Flow Risk Mapping Based on GIS and Extenics".
We have studied the valuable comments, and tried our best to revise the manuscript. The respond to the reviewer's comments are as follows:

Comment 1: The methodology to compute correlation factors and weights is not defined in an understandable way. The general concepts of extenics methods (and its advantages over other methods) has to be defined more precisely and the assossiated mathematical description has to be improved. The symbols used in the equations are not all defined and when they are, it is often without any explanation. Furthermore, the mathematical operation are not explained. For example in eq. 1, M, aj1-ajn and bj1-bjn are not defined, as the operation <a, b> and finally what does exactly means this kind of matrix [] (with columns of different size). Thus, this section is very confused and almost impossible to understand, that takes all credibility away from the results.
Reply: According to the reviewer's comment, we will compare and analyze the evaluation results of existing debris flow risk assessment methods in first section, the third paragraph. We will make clear statement about the advantages and disadvantages on the existing assessment methods, and give the summary about the difference between extenics approach and other methods on evaluating debris flow risk. At the same time, we will improve the interpretation of symbols in manuscript formulas.

Comment 2: The input variables ( for instance the historical data page 6, line 34 or rainfall page 7, line 11) are used to verify the data obtained. It is absolutely mandatory to find other parameters (or methods) than the input ones to check the results. Otherwise, they do not have any scientific validity. Moreover, to ensure the validity of the Class V, you claimed that the area (covered by the Class V) is constituted by (among other things) large relative elevation. However, when one looks at the debris flow risks as a function of the relative elevation, one sees that terrains with largest relative elevation fall in Class I and II. As far as I understand, there is a big conmtradiction here.
Reply: Thank you very much for the comment of the reviewer. I found that the data in the article were not introduced rigorously. The historical debris flow data (1981-1994) are used to calculate the correlation degree in this manuscript and the historical debris flow data (1995-2004) are used to verify the model. I will add the corresponding explanations in section 2.2 Data.

Comment 3: Some of your results are not intuitive, in this case a comment would be helpful. For example, risks become smaller with larger slope (page 5, lines 24-25). In this case, it is written: "For the valley with smaller slope, [...], smaller shear force means larger stability and lower probability of debris flow". It seems for me that your results are in complete disagreement with your interpretation, therefore an explication is expected and needed.
Reply: About the reviewer's comment, we will re-examine the conclusions in the manuscript and improve its logic and grammar. We will intuitively display the results of the analysis in the

main sentences of the paragraph.

Comment 4: Your study is based on field and historical data, which are not well described in the paper. It misses especially a discussion about the quality of the data and the associated uncertainty of measurement.

Reply: According to the reviewer's comment, we will add a detailed description of the different factor datasets in section 2.2, and discuss the accuracy of the dataset. And meanwhile, giving the reason why the dataset can be chosen.